# MedRAX: Medical Reasoning Agent for Chest X-ray

Adibvafa Fallahpour [* 1 2 3 4]   Jun Ma [* 2 3]   Alif Munim [* 3 5]   Hongwei Lyu [3]   Bo Wang [† 1 2 3 6]

## Abstract

Chest X-rays (CXRs) play an integral role in driving critical decisions in disease management and patient care. While recent innovations have led to specialized models for various CXR interpretation tasks, these solutions often operate in isolation, limiting their practical utility in clinical practice. We present MedRAX, the first versatile AI agent that seamlessly integrates state-of-the-art CXR analysis tools and multimodal large language models into a unified framework. MedRAX dynamically leverages these models to address complex medical queries without requiring additional training. To rigorously evaluate its capabilities, we introduce ChestAgentBench, a comprehensive benchmark containing 2,500 complex medical queries across 7 diverse categories. Our experiments demonstrate that MedRAX achieves state-of-the-art performance compared to both open-source and proprietary models, representing a significant step toward the practical deployment of automated CXR interpretation systems. Data and code have been publicly available at https://github.com/bowang-lab/MedRAX.

## 1. Introduction

Chest X-rays have been widely used to make critical decisions in disease detection, diagnosis, and monitoring, comprising the largest proportion of over 4.2 billion diagnostic radiology procedures performed annually worldwide (United Nations Scientific Committee on the Effects of Atomic Radiation, 2022). However, the systematic evaluation of key anatomical structures places a significant time burden on radiologists, requiring hours of careful analysis (Bahl et al., 2020; Adibi et al., 2025).

The gradual introduction of AI into clinical practice has demonstrated promising potential to alleviate this burden. Task-specific AI models have shown success in automating various aspects of CXR interpretation, from classification and segmentation to automated report generation (Yang et al., 2017; Huang et al., 2023; Tanno et al., 2024; Ouis & Akhloufi, 2024). When integrated into clinical workflows, these tools have improved report turnaround times and interobserver agreement (Baltruschat et al., 2021; Ahn et al., 2022; Pham et al., 2022; Shin et al., 2023). However, the fragmented nature of these solutions—each operating in isolation—has hindered their widespread adoption in practical clinical settings (Erdal et al., 2023; Fallahpour et al., 2024).

Foundation models (FMs), including large language models (LLMs) and large multimodal models, are promising solutions to this fragmentation, enabling unified, scalable AI-driven medical image-text reasoning. OpenAI's GPT-4 established dominance of this approach with unprecedented scale. Trained on enormous multimodal data, it demonstrated exceptional medical understanding and reasoning without explicit training (Nori et al., 2023; Yan et al., 2023; Javan et al., 2024; Eriksen et al., 2024; Baghbanzadeh et al., 2025). LLaVA-Med (Li et al., 2024b), trained on 15 million biomedical figure-caption pairs, established new benchmarks in medical visual question answering (VQA) with strong zero-shot image interpretation. CheXagent (Chen et al., 2024a) focused on CXR analysis, achieving GPT-4-level performance with significantly fewer parameters.

While FMs have advanced the field, they face critical limitations hindering direct clinical application. LMMs experience hallucinations and inconsistencies in reasoning, particularly concerning where medical accuracy is paramount. They struggle with complex multi-step reasoning required for diagnostic tasks, failing to systematically evaluate all relevant anatomical structures or integrate findings across different image regions. Their end-to-end architecture lacks the transparency and specialization of purpose-built medical AI tools. These limitations suggest a more structured, tool-based approach combining foundation model flexibility with clinical AI system reliability.

*Equal contribution [1]Department of Computer Science, University of Toronto, Toronto, Canada [2]Vector Institute, Toronto, Canada [3]University Health Network, Toronto, Canada [4]Cohere, Toronto, Canada [5]Cohere Labs, Toronto, Canada [6]Department of Laboratory Medicine and Pathobiology, University of Toronto, Toronto, Canada. Correspondence to: Adibvafa Fallahpour <adibvafa.fallahpour@mail.utoronto.ca>.

*Proceedings of the $42^{nd}$ International Conference on Machine Learning*, Vancouver, Canada. PMLR 267, 2025. Copyright 2025 by the author(s).

To bridge this gap, we present MedRAX, the first specialized AI agent framework for CXR interpretation. Our key contributions include:

- MedRAX, a specialized AI agent framework that seamlessly integrates multiple CXR analysis tools without additional training, dynamically orchestrating specialized components for complex medical queries.

- ChestAgentBench, a comprehensive evaluation framework with 2,500 complex medical queries across 7 categories, built from 675 expert-curated clinical cases to assess multi-step reasoning in CXR interpretation.

- Experiments show that MedRAX outperforms both general-purpose and biomedical specialist models, demonstrating substantial improvements in complex reasoning tasks while maintaining transparent workflows.

- Development of a user-friendly interface, enabling flexible deployment options from local to cloud-based solutions that address healthcare privacy requirements.

## 2. Related Work

### 2.1. LLM-based Agent Architectures

The emergence of AI agents built upon LLMs has fundamentally changed how we approach autonomous reasoning, planning, and tool use. Recent surveys on LLM-based agents (Xi et al., 2025; Zhao et al., 2023; Masterman et al., 2024) have outlined a generalizable agent framework comprising three core components: (1) a reasoning engine driven by LLMs, (2) perceptual modules that process multimodal inputs, and (3) action mechanisms that execute API calls, retrieve information, or interact with external tools.

This paradigm shift has enabled AI agents to surpass traditional task-specific models by dynamically adapting to diverse applications without additional training. However, despite these advances, there are very few LLM-based agents that have been evaluated for domain-specific robustness, particularly in high-stakes medical applications where hallucinations, lack of systematic reasoning, and specialized tool integration remain significant challenges.

### 2.2. Medical Agents

By enabling LMMs to operate in a collaborative, agentic setting, frameworks such as MDAgents (Kim et al., 2024) have demonstrated enhanced clinical reasoning through multi-agent interaction. Similarly, MMedAgent (Li et al., 2024a) explores tool integration across multiple medical imaging modalities, allowing LMMs to leverage external machine learning models for more robust decision-making.

However, MDAgents introduces significant computational overhead due to multi-agent coordination, while MMedA-gent's broad focus across imaging modalities may dilute its domain-specific expertise. Additionally, MMedAgent requires retraining to integrate new tools, reducing its flexibility for adapting to evolving clinical workflows.

More recently, o1-powered AI agents (Jaech et al., 2024) have been proposed as an alternative to traditional model-based approaches, demonstrating strong multi-step reasoning and improved diagnostic consistency. However, these systems also face critical challenges: (1) high computational demands, making them impractical for real-time applications, (2) closed-source and proprietary nature, limiting customization and adaptation to specific medical requirements, and (3) redundant reasoning in simpler tasks, leading to inefficiencies in tool selection and execution.

Alongside these agentic frameworks, specific large vision-language models like RaDialog (Pellegrini et al., 2025) have been developed for radiology report generation and conversational assistance. Furthermore, models such as M4CXR (Park et al., 2024) are exploring the multi-task potentials of multi-modal LLMs for chest X-ray interpretation.

### 2.3. Evaluation Frameworks

To systematically evaluate LLM-based agents, several benchmarks have been introduced. AgentBench (Liu et al., 2023) assesses multi-step reasoning, memory retention, tool use, task decomposition, and interactive problem-solving, revealing that even top-performing models like GPT-4o and Claude-3.5-Sonnet struggle with long-term context retention and autonomous decision-making. Expanding on these limitations, MMAU (Yin et al., 2024) evaluates agent capabilities across five domains—tool use, graph-based reasoning, data science, programming, and mathematics. Results highlight persistent weaknesses in structured reasoning and iterative refinement.

In software engineering, SWE-bench (Jimenez et al., 2023) presents 2,294 real-world GitHub issues to evaluate LLMs' ability to modify large codebases. By January 2025, the best-performing agent has solved less than 65% of issues, underscoring the challenges of multi-file reasoning and iterative debugging. These benchmarks collectively highlight LLM agents' deficiencies in contextual understanding, structured planning, and domain-specific tool use, reinforcing the need for specialized, clinically validated AI frameworks in high-stakes applications such as medical imaging.

Beyond general-purpose benchmarks, MedAgentBench (Jiang et al., 2025) assesses LLMs' ability to retrieve patient data, interact with clinical tools, and execute structured decision-making in interactive healthcare environments. Results indicate that even the best-performing model, GPT-4o, achieves only 72% accuracy, with substantial performance variability across different medical tasks.

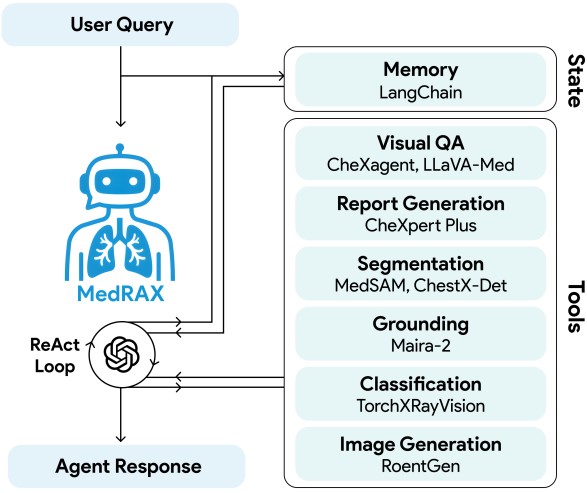

*Figure 1.* **Architecture of MedRAX.** The framework implements a ReAct loop that processes user queries by integrating short-term memory (LangChain) and specialized medical tools for visual QA (CheXagent (Chen et al., 2024b), LLaVA-Med (Li et al., 2024b)), segmentation (MedSAM (Ma et al., 2024; Kirillov et al., 2023; Ma et al., 2025), ChestX-Det (Lian et al., 2021; Zhao et al., 2017)), grounding (Maira-2 (Bannur et al., 2024)), report generation (model trained on CheXpert Plus (Irvin et al., 2019; Chambon et al., 2024)), classification (TorchXRayVision (Cohen et al., 2022; 2020)), and image generation (RoentGen (Chambon et al., 2022)).

**Algorithm 1** MedRAX ReAct Framework

**Input:**
$Q$: User query
$I$: Set of input CXR images (can be empty)
$T$: Available medical AI tools
$M$: Memory buffer
$t_{max}$: Maximum allowed time
**Output:**
$R$: Final response to query
**Initialize:**
$t_{start}$ = GetCurrentTime()
state = Observe($Q, I, M$)
**while** GetCurrentTime() $- t_{start} < t_{max}$ **do**
   thoughts = Reason($state, M$)
   **if** RequiresUserInput($thoughts$) **then**
      **return** GenerateUserPrompt($thoughts, M$)
   **end if**
   **if** CanGenerateResponse($thoughts$) **then**
      **return** GenerateResponse($thoughts, M$)
   **end if**
   tools = SelectTools($thoughts, T, M$)
   results = ExecuteParallel($tools, state$)
   $M = M \cup \{(thoughts, tools, results)\}$
   state = Observe($state, results, M$)
**end while**
**return** GenerateTimeoutResponse($state, M$)

AgentClinic (Schmidgall et al., 2024) offers a comprehensive multimodal agent benchmark to evaluate AI in simulated clinical environments, emphasizing interactive dialogue and active data collection processes. These findings reinforce the critical need for domain-specific benchmarks and highlight that while progress has been made, current medical agents still face significant challenges in tool integration, efficiency, and flexibility.

## 3. MedRAX

We present MedRAX, an open-source agent-based framework that can dynamically reason, plan, and execute multi-step CXR workflows. Compared to previous approaches (Chen et al., 2024b; Bansal et al., 2024), MedRAX integrates multimodal reasoning abilities with structured tool-based decision-making, allowing real-time CXR interpretation without unnecessary computational overhead. By balancing computational efficiency with domain specialization and eliminating the need for retraining when incorporating new tools, MedRAX offers greater adaptability to evolving clinical needs. Our framework integrates heterogeneous machine learning models—from lightweight classifiers to large LMMs—specialized for diverse downstream tasks, allowing it to decompose and solve complex medical queries by reasoning across multiple analytical skills (Figure 1).

### 3.1. LLM Driven Agent

MedRAX employs a LLM as the core to drive a ReAct (Reasoning and Acting) loop, which breaks down complex medical queries into sequential analytical steps (Yao et al., 2023). The system processes a user query through iterative cycles of (1) observation - analyzing the current state and query, (2) thought - determining required actions, and (3) action - executing relevant tools and integrating findings from previous steps to inform subsequent reasoning. Throughout this process, the system maintains a short-term memory of user interactions, tool outputs, and images to support multi-turn interactions. The reasoning loop continues until the system either generates a response or asks the user for additional input (Algorithm 1). Further details of the core methodology are provided in Appendix A.

### 3.2. Flexible Tool Integration

MedRAX integrates state-of-the-art models for various downstream CXR interpretation tasks:

- **Visual Question Answering (VQA).**
  Answering free-form questions about CXR images by combining visual understanding with medical knowledge.

  *Models*: CheXagent, a vision-language foundation model trained on CheXinstruct, with over 8.5M samples across

35 tasks, capable of fine-grained visual reasoning and CXR interpretation (Chen et al., 2024b).

LLava-Med, a biomedical 7B VLM, trained on 600K biomedical image-caption pairs from PMC-15M and 60K instruction-tuning data (Li et al., 2024b).

- **Segmentation**.
  Partitioning CXR images into semantically meaningful regions by assigning each region to anatomical structures.

  *Models*: MedSAM, a state-of-the-art biomedical segmentation model trained on 1,570,263 medical image-mask pairs, covering 10 imaging modalities and over 30 cancer types (Ma et al., 2024; 2025).

  PSPNet model trained on ChestX-Det dataset, consisting of 3,578 images from NIH ChestX-14, annotated with 13 common categories of diseases or abnormalities (Lian et al., 2021; Zhao et al., 2017).

- **Grounding**.
  Localizing specific visual regions in medical images that correspond to given textual descriptions or findings.

  *Model*: Maira-2, a 7B VLM trained on MIMIC-CXR, PadChest, and USMix datasets, excellent in grounding specific phrases or generating findings of a radiology report with or without grounding (Bannur et al., 2024).

- **Report Generation**.
  Writing radiology reports with findings and impressions.

  *Model*: A SwinV2 Transformer with a two-layer BERT decoder trained on 223K expert-annotated reports from CheXpert Plus dataset to generate findings and impressions (Irvin et al., 2019; Chambon et al., 2024).

- **Disease Classification**.
  Detecting and classifying pathologies and abnormalities.

  *Model*: A DenseNet-121 model from the TorchXRayVision library, trained on NIH ChestX-ray, CheXpert, MIMIC-CXR, and PadChest datasets. It can predict 18 pathology classes including Pneumonia, Pneumothorax, Edema, Effusion, and Nodule (Cohen et al., 2022; 2020).

- **Chest X-ray Generation**.
  Synthesizing realistic CXR images from text descriptions of anatomical features and pathologies.

  *Model*: RoentGen, a medical vision-language model adapted from Stable Diffusion, trained on the MIMIC-CXR dataset, generates diverse, high-fidelity chest X-rays given text prompts (Chambon et al., 2022).

- **Utilities**.
  Processing DICOM images, generating custom plots, and visualizing figures to user.

The agent continuously monitors tool outputs and errors, incorporating these results into its reasoning loop to inform

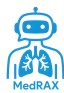
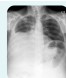

**User Query**
Ground the main disease in this CXR.

**MedRAX**
**<thought>** First, I need to diagnose the disease.
**<action>** {"tool": "chest_xray_classifier", "image_path": ...}

**<tool>** {"Effusion": 85%, "Atelectasis": 69%, ...}

**<thought>** The most probably disease is effusion. I will ground it.
**<action>** {"tool": "phrase_grounding", "phrase": "pleural effusion"}

**<tool>** {"bounding_boxes": [(0.05, 0.39, 0.42, 0.63)], ...}
**<tool>** {"tool": "image visualizer", "image_path": ...}

**<response>** CXR shows clear signs of pleural effusion. Here is a visualization of grounded result:

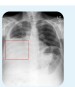

**User Query**
Segment the major affected lung.

**MedRAX**
**<observation>** Based on boundings, right lung is mostly affected.
**<thought>** I will segment the right lung.

**<action>** {"tool": "segmentation", "organs": ["right lung"], ...}

**<response>** Pleural effusion is mostly in the right lung. Here is the segmented region:

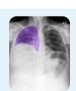

*Figure 2.* **MedRAX Interaction Flow.** An example of how MedRAX handles a multi-turn conversation through its ReAct loop (<thought>, <action>, <observation>) along with tool outputs and final response. For clarity, the production interface shows only tool outputs and agent responses.

subsequent tool selection. Through its memory, MedRAX caches tool outputs to prevent redundant computations, optimizing performance in multi-step analyses that might reference the same intermediate results.

The framework supports parallel execution of independent tools and flexible deployment configurations - tools can be quantized for efficiency and distributed across CPU or GPU. Figure 2 shows an example user interaction with MedRAX.

## 3.3. Modularity

MedRAX is built on the LangChain and LangGraph frameworks. The reasoning engine can be any LLM, accommodating both text-only and multimodal models, from open-source to proprietary. This flexibility enables deployments ranging from local installations to cloud-based solutions, addressing diverse healthcare privacy requirements. Our reference implementation uses GPT-4o with vision, while supporting integration of alternative models.

Each tool operates as an independent module with defined loading and inference. Tools can be modified, replaced, or repurposed for multiple tasks without affecting other components. Integration of new tools requires only a class definition specifying the tool's input/output formats and capabilities, with the LLM learning its usage without any training. The framework decouples tool creation from agent instantiation, enabling multiple agents to share tools and allowing each to access its own customized set of tools.

### 3.4. User-friendly Interface

MedRAX includes a production-ready interface built with Gradio that facilitates seamless deployment in clinical settings. The interface supports uploading of radiological images in all standard formats, including DICOM, and maintains an interactive chat session for natural multi-turn interactions. The interface further provides transparency into tool execution by tracking and displaying intermediate outputs. This end-to-end implementation enables quick integration of MedRAX into existing clinical workflows.

## 4. ChestAgentBench

While MedRAX offers a powerful framework for complex CXR interpretation, existing medical VQA benchmarks typically focus on simple, single-step reasoning tasks, making them insufficient for evaluating its full capabilities. In contrast, ChestAgentBench offers several distinctive advantages:

- It represents one of the largest medical VQA benchmarks, with 2,500 questions derived from expert-validated clinical cases, each with comprehensive radiological findings, detailed discussions, and multi-modal imaging data.

- The benchmark combines complex multi-step reasoning assessment with a structured six-choice format, enabling both rigorous evaluation of advanced reasoning capabilities and straightforward, reproducible evaluation.

- The benchmark features diverse questions across seven core competencies in CXR interpretation, requiring integration of multiple visual findings and reasoning to mirror the complexity of real-world clinical decision-making.

### 4.1. Dataset

We utilize Eurorad, the largest peer-reviewed radiological case report database maintained by the European Society of Radiology (ESR). This database contains detailed clinical cases consisting of patient histories, clinical presentations, and multi-modal imaging findings. Each case includes detailed radiological interpretations across different modalities, complemented by in-depth discussions that connect findings with clinical context, and concludes with reasoned interpretations, differential diagnosis list and a final diagnoses.

From its chest imaging section, we curated 675 patient cases with associated chest X-rays and complete clinical documentation. These cases cover 53 unique areas of interest including lung, thorax, and mediastinum. Figure 3 provides an overview of the benchmark, showing (a) the creation pipeline, (b) patient gender distribution, (c) age distribution, and (d) most frequent anatomical areas of interest. The dataset encompasses diverse clinical settings and a broad spectrum of pathologies, with detailed breakdowns provided in Appendix B.

### 4.2. Benchmark Creation

ChestAgentBench comprises six-choice questions, each designed to evaluate complex CXR interpretation capabilities.

We first established seven core competencies alongside reasoning that are essential for CXR interpretation:

- **Detection:** Identifying specific findings. (e.g., "Is there a nodule present in the right upper lobe?")

- **Classification:** Classifying specific findings. (e.g., "Is this mass benign or malignant in appearance?")

- **Localization:** Precise positioning of findings. (e.g., "In which bronchopulmonary segment is the mass located?")

- **Comparison:** Analyzing relative sizes and positions. (e.g., "How has the pleural effusion volume changed compared to prior imaging?")

- **Relationship:** Understanding relationship of findings. (e.g., "Does the mediastinal lymphadenopathy correlate with the lung mass?")

- **Diagnosis:** Interpreting findings for clinical decisions. (e.g., "Given the CXR, what is the likely diagnosis?")

- **Characterization:** Describing specific finding attributes. (e.g., "What are the margins of the nodule - smooth, spiculated, or irregular?")

- **Reasoning:** Explaining medical rationale and thought. (e.g., "Why do these findings suggest infectious rather than malignant etiology?")

These competencies are combined into five question types, each designed to evaluate specific combinations of core competencies while requiring medical reasoning:

- **Detailed Finding Analysis:** detection, localization, and characterization

- **Pattern Recognition & Relations:** detection, classification, and relationships

- **Spatial Understanding:** localization, comparison, and relationships

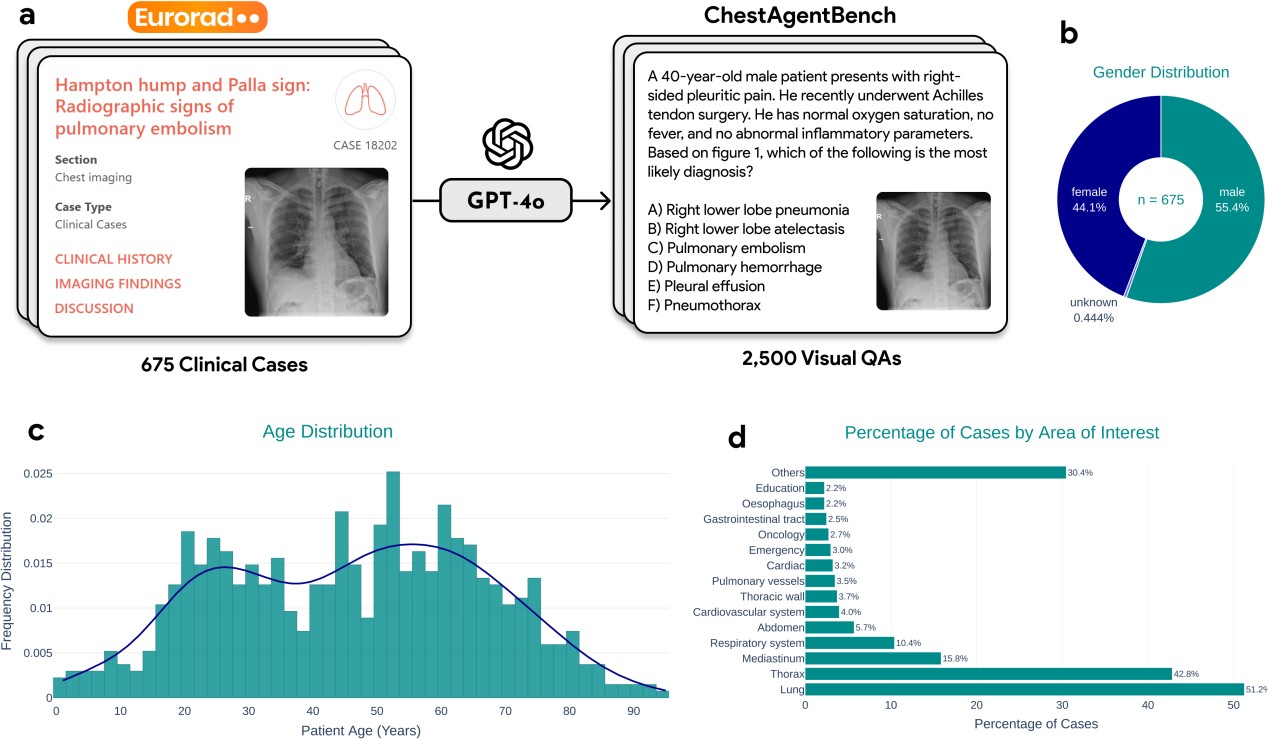

*Figure 3.* **Overview of ChestAgentBench.** (a) Benchmark creation pipeline that uses GPT-4o to generate 2,500 six-choice questions from 675 Eurorad clinical cases. (b) Gender distribution, showing 55.4% male, 44.1% female, and 0.45% unknown. (c) Age distribution, a bimodal with a mean age of 46.0 years (SD=20.4, median=47.0 years). (d) Distribution of anatomical areas of interest across cases, with lung (51.2%), thorax (42.8%), and mediastinum (15.8%) representing the most frequently examined regions from 53 unique areas.

- **Clinical Decision Making:** classification, comparison, and diagnosis

- **Diagnostic Characterization:** classification, characterization, and diagnosis

For each clinical case and question type, we first prompted GPT-4o to analyze the case and generate a six-choice question that would best assess the target analytical skills of that question type. We then instructed it to ensure the question has the necessary context from the clinical case and its correct answer could be explicitly verified from the case's radiological findings and discussion.

We performed automated quality verification using GPT-4o to evaluate each question for: (1) structural consistency (six-choice format with one correct answer), (2) explicit grounding in the clinical and radiological context, and (3) clear verifiability from the original Eurorad case material. Questions failing these criteria were automatically excluded.

The benchmark uses a straightforward accuracy metric (percentage of correct answers) to enable easy evaluation across different agent architectures. All questions underwent quality check, during which we removed questions that exhibited issues such as ungrounded answers or missing information.

## 5. Experiments

### 5.1. Implementations

MedRAX uses GPT-4o as its backbone LLM, and we deploy it on a single NVIDIA RTX 6000 GPU using the same configuration as described in Section 3. It integrates CheXagent (Chen et al., 2024b) and LLaVA-Med (Li et al., 2024b) for visual QA, Maira-2 for grounding (Bannur et al., 2024), a model trained on ChestX-Det for segmentation (Lian et al., 2021), TorchXRayVision for classification (Cohen et al., 2022), and a model trained on CheXpert Plus for report generation (Chambon et al., 2024).

MedRAX implements tool execution with structured JSON API calls, where the agent formulates precise requests with required arguments (e.g., image paths, text prompts) to call target tools. We evaluate all baseline models using their official implementations and recommended configurations.

We process model responses using regex to extract letter choices. For unclear responses, errors, or timeouts, we retry up to three times. Responses that remain invalid or do not choose a single choice are marked incorrect.

*Table 1.* Model Performance on ChestAgentBench. Accuracy (%) of five vision-language models (LLaVA-Med (Li et al., 2024b), CheXagent (Chen et al., 2024a), Llama-3.2-90B, GPT-4o, and MedRAX) compared across seven categories of our 2,500-question benchmark. MedRAX significantly outperforms both general-purpose models and specialized biomedical models across all categories.

| Categories | LLaVA-Med | CheXagent | Llama-3.2-90B | GPT-4o | MedRAX |
|---|---|---|---|---|---|
| Detection | 32.4 | 38.7 | 58.1 | 58.7 | **64.1** |
| Classification | 30.8 | 34.7 | 56.5 | 54.6 | **62.9** |
| Localization | 30.2 | 42.5 | 59.9 | 59.0 | **63.6** |
| Comparison | 30.6 | 38.5 | 57.5 | 55.5 | **61.8** |
| Relationship | 31.8 | 39.8 | 59.3 | 59.0 | **63.1** |
| Diagnosis | 29.3 | 33.5 | 55.9 | 52.6 | **62.5** |
| Characterization | 28.8 | 34.2 | 58.0 | 56.1 | **64.0** |
| **Overall** | 28.7 | 39.5 | 57.9 | 56.4 | **63.1** |

*Table 2.* Model Performance on CheXbench. Accuracy (%) of five vision-language models (LLaVA-Med (Li et al., 2024b), CheXagent (Chen et al., 2024a), Llama-3.2-90B, GPT-4o, and MedRAX) compared on 238 Visual QA (Rad-Restruct and SLAKE) and 380 Image-Text Reasoning questions (OpenI). MedRAX excels in VQA while achieving the best overall performance.

| Categories | LLaVA-Med | CheXagent | Llama-3.2-90B | GPT-4o | MedRAX |
|---|---|---|---|---|---|
| Visual QA | | | | | |
| Rad-Restruct | 34.9 | 57.1 | 62.6 | 53.9 | **68.7** |
| SLAKE | 55.5 | 78.1 | 74.0 | **85.4** | 82.9 |
| Fine-Grained Reasoning | 45.8 | **59.0** | 49.2 | 51.1 | 52.6 |
| **Overall** | 45.4 | 64.7 | 61.9 | 63.5 | **68.1** |

### 5.2. Experimental Setup

We evaluate MedRAX against four core models: LLaVA-Med, a finetuned LLaVA-13B model for biomedical visual question answering (Li et al., 2024b), CheXagent, a Vicuna-13B VLM trained for CXR interpretation (Chen et al., 2024b), along with GPT-4o and Llama-3.2-90B Vision as popular closed and open-source multimodal LLMs respectively. Additionally, we compare against baseline models from recent literature including RadFM (Wu et al., 2023), a generalist foundation model for radiology, MAIRA-1 (Hyland et al., 2024), a specialized multimodal model for radiology report generation, LLaVA-Rad (Zambrano Chaves et al., 2024), a radiology-adapted version of LLaVA, and Med-PaLM M 84B (Tu et al., 2023), a large-scale biomedical foundation model, with performance values obtained from M4CXR paper (Park et al., 2024).

We evaluate models on four complementary benchmarks:

(1) **ChestAgentBench**, our proposed benchmark described in Section 4, which assesses comprehensive CXR reasoning through 2,500 six-choice questions across seven categories: detection, classification, localization, comparison, relationship, characterization, and diagnosis. Model performance is measured by accuracy across all questions.

(2) **CheXbench**, a popular benchmark that evaluates seven clinically-relevant CXR interpretation tasks. We specifically focus on the visual question answering (238 questions from Rad-Restruct (Pellegrini et al., 2023) and SLAKE (Liu et al., 2021) datasets) and fine-grained image-text reasoning (380 questions from OpenI dataset) subsets, as they most closely mirror complex clinical workflows that require precise differentiation between similar findings.

(3) **MIMIC-CXR Radiology Report Generation**, which evaluates single-image chest X-ray findings generation of 3,858 images from the MIMIC-CXR test set. This benchmark assesses clinical accuracy of generated reports using micro-averaged F1 scores (mF1-14, mF1-5) and macro-averaged F1 scores (MF1-14, MF1-5) for 14 medical observation labels and 5 key findings (cardiomegaly, edema, consolidation, atelectasis, pleural effusion), respectively.

(4) **SLAKE VQA**, which evaluates medical visual question answering using 114 chest X-ray test samples with close-ended questions in English, filtered from the original SLAKE test set of 2,094 samples. Performance is measured by accuracy (exact matches) and recall (proportion of ground truth words present in generated responses).

For more information on preparation and evaluation on benchmarks 3 and 4, see M4CXR paper (Park et al., 2024).

*Table 3.* Single-image performance on MIMIC-CXR test set. Clinical accuracy (%) of generated radiology findings, evaluated using CheXbert F1 scores for 14 medical observations. mF1-14/mF1-5: micro-averaged F1 scores for all 14/5 key labels (cardiomegaly, edema, consolidation, atelectasis, pleural effusion); MF1-14/MF1-5: macro-averaged F1 scores for all 14/5 key labels. MedRAX is compared with CheXagent (Chen et al., 2024b), MAIRA-1 (Hyland et al., 2024), LLaVA-Rad (Zambrano Chaves et al., 2024), Med-PaLM M 84B (Tu et al., 2023), and M4CXR (Park et al., 2024). Performance values for baseline models obtained from Park et al. (Park et al., 2024).

| Model | mF1-14 | mF1-5 | MF1-14 | MF1-5 |
|---|---|---|---|---|
| Med-PaLM M 84B | 53.6 | 57.9 | 39.8 | **51.6** |
| CheXagent | 39.3 | 41.2 | 24.7 | 34.5 |
| MAIRA-1 | 55.7 | 56.0 | 38.6 | 47.7 |
| LLaVA-Rad | 57.3 | 57.4 | 39.5 | 47.7 |
| M4CXR | 60.6 | 61.8 | **40.0** | 49.5 |
| **MedRAX** | **79.1** | **64.9** | 34.2 | 48.2 |

*Table 4.* Medical VQA performance on SLAKE benchmark. Accuracy (%) and recall (%) of vision-language models on 114 chest X-ray visual question answering samples with close-ended English questions. MedRAX achieves state-of-the-art performance compared to RadFM (Wu et al., 2023), CheXagent (Chen et al., 2024b), and M4CXR (Park et al., 2024). Performance values for baseline models obtained from Park et al. (Park et al., 2024).

| Model | Accuracy | Recall |
|---|---|---|
| RadFM | 68.4 | 69.7 |
| CheXagent | 71.1 | 73.2 |
| M4CXR | 85.1 | 86.0 |
| **MedRAX** | **90.35** | **91.23** |

### 5.3. Quantitative Analysis

**ChestAgentBench.** Shown in Table 1, MedRAX achieves consistently state-of-the-art performance (63%) across all seven categories, a significant improvement over the baseline models. There is a clear performance hierarchy among models, with GPT-4o (56.4%) and Llama-3.2-90B (57.9%) performing notably better than specialized medical models like CheXagent (39.5%) (Chen et al., 2024a) and LLaVA-Med (28.7%) (Li et al., 2024b). Interestingly, general-purpose VLMs outperform domain-specific ones across all categories, with particularly large gaps in characterization and diagnosis tasks.

**CheXbench.** Shown in Table 2, we observe distinct performance patterns across different task types. On visual QA tasks, MedRAX demonstrates strong performance on Rad-Restruct (68.7%) and SLAKE (82.9%). This notably surpasses both domain-specific CheXagent (57.1%, 78.1%) (Chen et al., 2024a) and larger general-purpose models like GPT-4o (53.9%, 85.4%), suggesting that our tool-based approach particularly excels at fine-grained visual understanding. However, on image-text reasoning tasks, we observe a

significant performance drop across all models, with even the best-performing CheXagent achieving only 59.0% accuracy, almost equal to random performance (50% baseline).

**MIMIC-CXR Radiology Report Generation.** Shown in Table 3, MedRAX achieves the highest micro-averaged F1 scores with mF1-14 of 79.1% and mF1-5 of 64.9%, substantially outperforming the next-best M4CXR (60.6%, 61.8%) and other baselines. However, MedRAX shows lower macro-averaged performance, with M4CXR achieving the highest MF1-14 of 40.0% and Med-PaLM M 84B leading MF1-5 at 51.6%. This suggests MedRAX's stronger performance on more prevalent conditions.

**SLAKE VQA.** Shown in Table 4, MedRAX demonstrates clear superiority in medical visual question answering, achieving 90.35% accuracy and 91.23% recall. This represents significant improvements over M4CXR (85.1% accuracy, 86.0% recall) and CheXagent (71.1% accuracy, 73.2% recall). The consistent performance gains across both metrics indicate MedRAX's tool-based approach effectively handles diverse medical VQA tasks while maintaining high precision in response generation.

### 5.4. Case Studies

We present two representative cases that compare MedRAX to GPT-4o (Figure 5.3).

**Medical Device Identification (Eurorad Case 17576).** This question asks the model to determine the type of tube present in the CXR. GPT-4o incorrectly suggests an endotracheal tube based on the central positing of the tube alone. MedRAX, integrated findings from multiple tools like report generation and visual QA, and correctly identifies a chest tube despite one tool (LLaVA-Med (Li et al., 2024b)) suggesting otherwise. This demonstrates MedRAX's ability to resolve conflicting tool outputs through systematic reasoning.

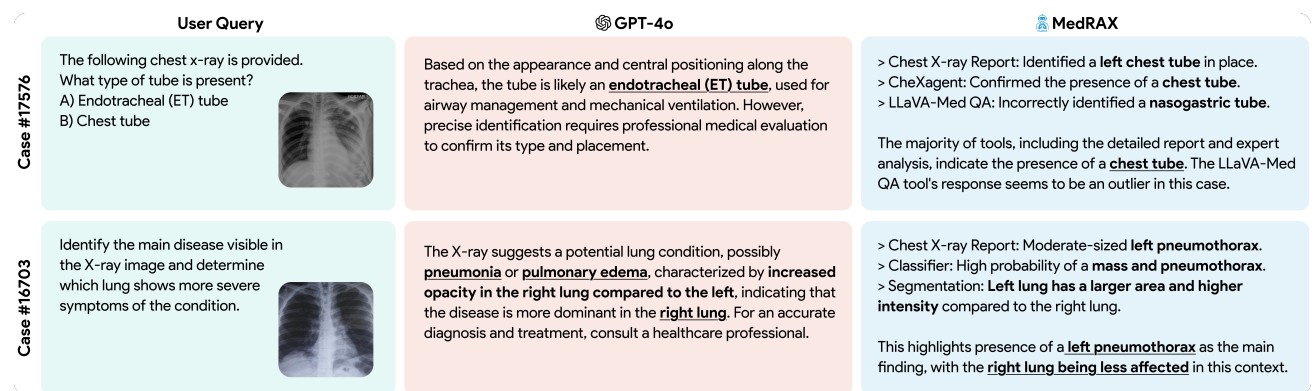

*Figure 4.* **MedRAX and GPT-4o Case Study.** (Case 17576) Correct answer is chest tube. GPT-4o incorrectly identifies as endotracheal tube based on position, while MedRAX correctly identifies chest tube by integrating multiple tool outputs, even resolving conflicting tool suggestions. (Case 16703) Correct answer is left pneumothorax. GPT-4o misdiagnoses as right-sided pneumonia/edema, while MedRAX correctly identifies left pneumothorax through sequential tool application for disease detection and comparative lung analysis.

**Multi-step Disease Diagnosis (Eurorad Case 16703).** This questions asks about diagnosing and comparing disease severity across lungs. GPT-4o misinterprets the CXR as showing pneumonia with right lung predominance. MedRAX, through sequential tool application of report generation for disease identification and segmentation for lung opacity analysis, correctly determines left pneumothorax as the main finding. This demonstrates MedRAX's ability to break down complex queries into targeted analytical steps.

## 6. Discussion

MedRAX achieves state-of-the-art performance in complex CXR interpretation tasks, outperforming both general-purpose and specialized medical models. We discover valuable insights about structured tool use in medical AI, suggesting that a hybrid approach—leveraging both large-scale reasoning capabilities and domain-specific expertise—offers superior performance over purely end-to-end models.

**Task Decomposition**. MedRAX shows that the ReAct loop dynamically composes complex reasoning chains while maintaining computational efficiency. The performance gap suggests that explicit decomposition provides advantages that scale alone cannot achieve. The process produces clear decision traces, enhancing transparency with implications beyond medical imaging for model-tool integration.

**Generalists Versus Specialists**. A key insight is the superior performance of general-purpose models (GPT-4o, Llama-3.2-90B) over specialized medical models (LLaVA-Med (Li et al., 2024b), CheXagent (Chen et al., 2024a)). This suggests medical specialization may sacrifice broader reasoning capabilities from large-scale pretraining. MedRAX bridges this gap by integrating domain-specific tools while maintaining generalist reasoning.

**Limitations.** While MedRAX excels in structured reasoning, it sometimes struggles with resolving contradictory tool outputs, particularly in fine-grained visual tasks when classification and segmentation tools provide conflicting interpretations. Additionally, the system's computational overhead from running multiple specialized tools can impact response times compared to end-to-end models. The framework also lacks robust uncertainty quantification mechanisms.

**Future Work.** Key areas for advancement include formally analyzing optimal tool utilization balance, as neither full reliance nor complete absence yielded best results. This involves exploring trade-offs between accuracy gains and computational costs, speed, error propagation, and spurious correlations in pretrained models. Systematic investigation of prompting techniques for critical tool evaluation is needed to optimize LLM-tool interactions. Reinforcement learning approaches could enhance reasoning capabilities and reduce hallucinations, following advances in models like DeepSeek-R1 (Guo et al., 2025). Comprehensive clinical validation remains essential for establishing real-world utility.

## 7. Conclusion

MedRAX establishes a new benchmark in AI-driven CXR interpretation by integrating structured tool orchestration with large-scale reasoning. Our evaluation demonstrates its superiority over both general-purpose and domain-specific models, reinforcing the advantages of explicit stepwise reasoning in medical AI. These findings highlight the potential of combining foundation models with specialized tools, a principle applicable to broader healthcare domains. Future work should focus on optimizing tool selection, uncertainty-aware reasoning, and expanding capabilities to multimodal medical imaging for greater clinical impact.

## Impact Statement

MedRAX aims to improve chest X-ray interpretation, enhancing diagnostic accuracy and efficiency as an AI co-pilot designed to augment, not replace, clinical expertise. We acknowledge AI risks, including model bias, hallucinations, and data privacy, and emphasize the need for robust validation before any clinical deployment to ensure patient safety.

Our development utilized the de-identified Eurorad database, and its use, along with ChestAgentBench creation, strictly adheres to the CC BY-NC-SA 4.0 license; ChestAgentBench will be released under the same terms. To address data privacy, especially for datasets like MIMIC-CXR requiring stricter handling, we employed the Azure OpenAI Service configured to opt-out of data logging, following Physionet recommendations.

MedRAX also supports local LLM deployment for maximum data control. Our public code release will include guidelines for these privacy-preserving configurations, underscoring our commitment to responsible AI integration in healthcare.

## Acknowledgements

We thank Mohammed Baharoon for their assistance in preprocessing the Eurorad dataset. We are particularly grateful to Karan Singhal and Shekoofeh Azizi for their thorough feedback on ChestAgentBench's design, which significantly improved the benchmark's clinical relevance and evaluation methodology. We also thank Yubin Kim for their insightful perspectives on medical AI agent architectures. This work was supported by the University of Toronto, University Health Network (UHN) and Vector Institute.

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

# A. MedRAX Core Methodology

This appendix provides detailed exposition of Algorithm 1, which forms the core operational framework of MedRAX. The algorithm implements a ReAct (Reasoning and Acting) loop that enables dynamic tool orchestration for complex CXR interpretation tasks through iterative cycles of reasoning, acting, and observing.

## A.1. The ReAct Cycle: Foundational Concept

MedRAX operates on a cyclical Reason-Act (ReAct) principle, distinguishing it from traditional one-shot input-output models. The agent iterates through three core phases:

- **Reason**: Analyze the current situation (user query, conversation history, previous results) and decide on the next step
- **Act**: If necessary, perform an action, typically using a specialized tool to gather more information or perform a specific task
- **Observe**: Incorporate the results of the action back into its understanding of the situation and repeat the cycle

This iterative process enables the agent to tackle complex problems requiring multiple steps or external information sources, such as analyzing CXRs using various diagnostic tools sequentially or in parallel.

## A.2. Essential Components and Architecture

MedRAX relies on four interconnected components to execute the ReAct cycle:

### A.2.1. CORE REASONING ENGINE (LLM)

A powerful multimodal Large Language Model (e.g., GPT-4o) capable of understanding text, images, and crucially, using tools. The LLM performs the "Reason" step by analyzing the situation and deciding whether to answer directly or request tool execution. The chosen LLM must demonstrate proven multimodal understanding and reliable tool-calling adherence.

MedRAX employs the following system prompt to guide the reasoning engine:

> *You are an expert medical AI assistant who can answer any medical questions and analyze medical images similar to a doctor. Solve using your own vision and reasoning and use tools to complement your reasoning. Make multiple tool calls in parallel or sequence as needed for comprehensive answers. Critically think about and criticize the tool outputs. If you need to look up some information before asking a follow up question, you are allowed to do that.*

### A.2.2. SPECIALIZED TOOLBOX

A collection of pre-trained AI models for specific CXR tasks including classification, segmentation, VQA, grounding, and report generation. The framework encapsulates each specialized model in a software interface that enables LLM interaction through standardized tool wrappers.

**Tool Interface (Wrappers)**: Each tool wrapper defines four critical components:

- **name**: A unique identifier (e.g., "cxr_classifier")
- **description**: A natural language explanation detailing what the tool does, its inputs, and outputs (e.g., "Takes a CXR image reference and returns likely pathologies and their probabilities"). This description acts as implicit prompting, guiding the LLM on when to use the tool
- **input_schema**: A structured definition of required inputs (e.g., image identifier)
- **execution_logic**: Code to run the tool with given inputs and return results

The description field within each wrapper is paramount—it serves as the primary mechanism for the LLM to understand tool capabilities and appropriate usage contexts. Detailed tool descriptions used by MedRAX to understand utility and calling procedures are available in our public GitHub repository.

A.2.3. WORKFLOW ORCHESTRATOR

Manages the overall ReAct cycle, directing information flow between the LLM, Toolbox, and Memory. Key functions include:

- Calling the LLM for the "Reason" step

- Parsing LLM responses to check for tool requests

- Calling appropriate tools from the Toolbox for the "Act" step

- Updating Memory with LLM responses and tool results

- Deciding when the cycle is complete

A.2.4. AGENT MEMORY (STATE)

Stores the complete interaction history as a sequence of structured messages including user input, LLM thoughts/responses, and tool results. The memory provides necessary context for LLM reasoning at each step, ensuring the agent "remembers" previous findings and interactions throughout multi-turn conversations.

## A.3. Detailed Algorithm Execution Flow

A.3.1. STEP 0: INITIALIZATION AND STATE PREPARATION

The `Observe(Q, I, M)` function consolidates three critical information sources: the user query ($Q$), input CXR images ($I$), and the agent's memory buffer ($M$). The predefined system prompt is loaded to establish the agent's role and reasoning approach.

Image data is represented within Memory/Messages using URIs, IDs, or embedded data like base64. The chosen LLM must process these image representations alongside text, while tool wrappers operating on images must access image data based on references passed in their arguments.

A.3.2. STEP 1: LLM-DRIVEN REASONING PROCESS

**Prompt Construction**: The Orchestrator prepares comprehensive input for the LLM including:

- The system prompt defining the agent's role and approach

- The entire message history from Memory, formatted chronologically

- Tool definitions: name, description, and input_schema for all available tools, formatted according to the LLM provider's API requirements

**LLM Decision Process**: The `Reason(state, M)` function generates structured thoughts through comprehensive analysis. The LLM's decision to respond directly or use tools stems from training on vast datasets including instruction following and tool-use examples, combined with specific context. The LLM analyzes:

- Whether current message history and internal knowledge sufficiently answer the query according to system prompt instructions

- Available tools (via descriptions) to determine if any tool can provide more accurate or efficient information than internal knowledge (e.g., precise classification probabilities, segmenting specific regions)

- Whether to ask clarifying questions if the query is ambiguous or lacks information

**Response Generation**: Based on its decision, the LLM can:

- **Answer/Clarify Directly**: Generate only text response (final answer or clarifying question back to user)

- **Use Tools**: Generate structured Tool Call Request(s), potentially requesting multiple different tools or the same tool multiple times with different arguments for parallel information gathering

### A.3.3. STEP 2: CONDITIONAL DECISION MAKING

The Orchestrator examines the LLM's response to determine the next action:

**User Input Requirements**: The `RequiresUserInput(thoughts)` condition is implicitly evaluated during reasoning. When the LLM's thoughts indicate ambiguity or insufficient information that cannot be resolved by tools, `GenerateUserPrompt(thoughts, M)` formulates a natural language question based on its thoughts and context in memory to elicit needed information.

**Tool Execution Path**: If Tool Call Requests are present, the system proceeds to tool execution. If absent, the system routes to response generation where the LLM synthesizes the final answer.

### A.3.4. STEP 3: DYNAMIC TOOL SELECTION AND EXECUTION

**Tool Processing**: The `SelectTools(thoughts, T, M)` and `ExecuteParallel(tools, state)` functions handle each Tool Call Request by processing them potentially concurrently if designed for parallelism:

- **Identify Tool**: Look up requested tool_name in Toolbox to find corresponding Tool Wrapper object/class
- **Validate & Extract Arguments**: Parse arguments dictionary provided by LLM for that specific call, checking if arguments match the tool's defined input_schema. If validation fails, an error is generated
- **Invoke Tool**: If arguments are valid, call the specific execution function/method within the Tool Wrapper, which contains actual code to interact with the specialized model (load model, preprocess input, run inference, postprocess output)
- **Receive Result**: Capture return value from tool's execution logic (typically string or structured JSON data representing findings or error message if execution failed)
- **Format Result**: Package result into structured Tool Result Message including result content and unique ID corresponding to the Tool Call Request

The system supports parallel execution of independent tools while providing flexible deployment configurations—tools can be quantized for efficiency and distributed across CPU or GPU resources.

### A.3.5. STEP 4: MEMORY MANAGEMENT AND LOOP CONTROL

**Memory Update**: The operation $M = M \cup \{(thoughts, tools, results)\}$ maintains comprehensive records of the agent's reasoning process. This critical step logs the LLM's thoughts that led to tool calls, the identity of tools used, and results obtained. This persistent memory serves multiple functions:

- Provides context for subsequent reasoning cycles
- Enables reference to previous findings when synthesizing complex analyses
- Supports multi-turn conversations by preserving interaction history
- Caches tool outputs to prevent redundant computations in multi-step analyses that might reference the same intermediate results

**Loop Control**: If coming from tool execution, the system loops back to Step 1 (Reasoning) where the LLM receives updated Memory including Tool Result Messages. The LLM can now observe and rethink, processing tool outputs by:

- Synthesizing tool results with previous context
- Correcting understanding if tools returned unexpected information
- Deciding if more reasoning or different tool calls are needed
- Formulating final answers based on gathered evidence

When `CanGenerateResponse(thoughts)` evaluates to true, the LLM synthesizes the final response drawing upon concluding thoughts and information in memory.

### A.4. Error Handling and Robustness Mechanisms

MedRAX incorporates several critical mechanisms to ensure robust operation:

**Timeout Management**: The $t_{max}$ parameter enforces maximum execution time, preventing indefinite loops while allowing sufficient time for complex multi-step analyses via `GenerateTimeoutResponse(state, M)`.

**Tool Failure Recovery**: When tool execution fails, error messages are formatted as Tool Result Messages clearly indicating the error. When the flow loops back to reasoning, the LLM sees this error message and can decide how to proceed based on its instructions (e.g., try different tool, ask user for clarification, inform user of limitations).

### A.5. Implementation Architecture

**Node and Edge Structure**: The workflow orchestrator is implemented using libraries like LangGraph that provide structures for state machine implementation:

- **Nodes (Processing Steps)**: Python functions or class methods representing processing steps (ReasoningStep, ActionStep). Each function accepts current agent Memory/State object as input, performs specific logic, and returns dictionary containing updates to Memory/State
- **Standard Edges**: Unconditional transitions configured by stating flow between nodes
- **Conditional Edges**: Decision points implemented with routing functions that take current Memory/State as input, apply logic, and return string indicating next node name to execute

This comprehensive methodology establishes MedRAX as a flexible, robust, and transparent framework for AI-assisted CXR interpretation, capable of dynamic adaptation to diverse clinical scenarios while maintaining clear decision traces throughout the reasoning process.

## B. ChestAgentBench: Comprehensive Benchmark Statistics

### B.1. Clinical Setting Distribution

ChestAgentBench's 675 cases represent diverse clinical environments:

- Emergency Room (ER): 19.7% (133 cases)
- Intensive Care Unit (ICU): 4.9% (33 cases)
- Other hospital settings: 75.4% (509 cases)

### B.2. Pathology Distribution

The benchmark encompasses 20 distinct pathological findings with frequencies ranging from 26.3% (Mass) to 1.5% (Cardiomegaly), as detailed in Table 5. This distribution reflects real-world clinical prevalence patterns, with structural abnormalities (masses, nodules) and fluid collections (effusions) representing the most common findings, while cardiac and infectious conditions appear less frequently.

*Table 5.* Complete Distribution of various Chest X-ray Findings in ChestAgentBench (Frequency % across 675 clinical cases)

| Finding | (%) | Finding | (%) |
|---|---|---|---|
| Mass | 26.3 | Interstitial findings | 6.9 |
| Effusion | 24.6 | Bronchiectasis | 5.9 |
| Pleural Effusion | 21.5 | Atelectasis | 4.9 |
| Consolidation | 21.3 | Fibrosis | 4.1 |
| Nodule | 17.2 | Edema | 3.9 |
| Calcification | 10.7 | Cavitation | 3.9 |
| Pneumothorax | 7.6 | Fracture | 3.0 |
| Lymphadenopathy | 7.6 | Tuberculosis | 2.6 |
| Pneumonia | 7.2 | Metastasis | 2.6 |
| Emphysema | 7.1 | Cardiomegaly | 1.5 |

