# OpenReview forum: "MedRAX: Medical Reasoning Agent for Chest X-ray"
_ICML.cc/2025/Conference — ICML 2025 poster_

### Official Review · Reviewer_4LiS · 2025-03-13

**Overall Recommendation:** 2

**Summary:**

This paper introduces an AI agent multimodal, multi-task chest X-ray (CXR) interpretation and analysis called MedRAX. This method leverages an LLM-controlled reasoning and acting (ReAct) loop, dynamically reasoning through multi-step queries and selecting pretrained, task-specific tools to complete each step. The authors also introduce a new benchmark of 2,500 medical queries, on which MedRAX outperforms existing generalist and domain-specific multimodal language models on a wide variety of tasks.

## Update after rebuttal
I thank the authors for their detailed rebuttal, particularly the inclusion of quantitative results on existing benchmarks. While I appreciate the additional descriptions of methodology, this still does not meet the standard of reproducibility. I understand that the open-source implementation will facilitate reproducibility in practice, but it is critical for a scientific paper to communicate the methodology clearly so that reviewers and readers can understand and evaluate its validity. As much as I appreciate the ambition and forward-thinking nature of this paper, I feel that the technical description of methodology remains a major shortcoming -- I am left wondering what is happening "under the hood" at each stage beyond the high-level descriptions provided. I will maintain my original recommendation of Weak Reject.

**Claims And Evidence:**

If the goal is to claim superiority over existing state-of-the-art, then experimental validation appears sound but could be strengthened. The justification for creating a new benchmark was not entirely made clear – what specifically is lacking in existing medical (or CXR-specific) reasoning and analysis benchmarks?

Further, for a model with such diverse capabilities, it is surprising to see performance reported on two benchmark datasets against just four baseline models. I imagine this is due to the limited number of models capable of all tasks, but analysis could be performed on subsets of tasks or benchmarks could be chosen to facilitate comparison with more existing methods.

**Essential References Not Discussed:**

This is a nascent space, and the authors sufficiently capture the few relevant prior studies. Perhaps AgentClinic [1] is worth mentioning, even though fully text-based, as an example of “agentic” AI for clinical use cases.

While M4CXR [2] does not adopt an agentic approach, it is a relevant example of a multimodal foundation model for CXR analysis capable of a wide variety of tasks like MedRAX.

[1] Schmidgall, Samuel, et al. "AgentClinic: a multimodal agent benchmark to evaluate AI in simulated clinical environments." arXiv preprint arXiv:2405.07960 (2024).

[2] Park, Jonggwon, et al. "M4CXR: Exploring Multi-task Potentials of Multi-modal Large Language Models for Chest X-ray Interpretation." arXiv preprint arXiv:2408.16213 (2024).

**Experimental Designs Or Analyses:**

Experimental design appears sound, but it is difficult to evaluate such complex multi-task evaluation without more granular methods description or access to source code. Key methodological and implementation details range from completely missing to insufficiently described.

**Methods And Evaluation Criteria:**

The benchmark datasets used in this paper are appropriate for this unique and diverse multi-task setting. The proposed method is uniquely tailored to handling the suite of CXR analysis tasks considered.

**Other Comments Or Suggestions:**

This is a difficult paper to evaluate as a reviewer. I cannot in good conscience recommend acceptance for a paper that leaves so many core methodological details undescribed. However, if these details are clarified in the rebuttal (and, ideally, if evaluation is strengthened), then this is a very straightforward acceptance of what I could imagine becoming an influential paper.

Minor comments:
-	Line 157 on RHS: Change “Llava-Med” -> “LLava-Med”
-	I would clarify the metric used in each table caption, even if it is mentioned elsewhere in the text.

**Other Strengths And Weaknesses:**

*Strengths*:
- Highly unique and novel idea that is likely to garner significant attention and even be a cornerstone for future work. It is a forward-thinking approach to CXR interpretation
- The paper is well-written and clearly organized with simple, effective visuals.


*Weaknesses*:
- This reads more like a tech report than a scientific paper – there is virtually no description of the core methodology. Nearly every line in Algorithm 1 requires concrete definition and detailed explanation. Certain key components of this should appear in the main text, but most implementation details should minimally appear in the Supplement (currently there is none!).
- While I appreciate the work that went into curating a new benchmark, the justification for this needs to be clarified – what specifically is lacking about current benchmarks?
- Similarly, while I am aware that only a select few models are even capable of all tasks considered in this setting, evaluation can be expanded to more benchmarks and more baseline methods. E.g., using established baselines could facilitate direct comparison with existing state-of-the-art on individual tasks or subsets of tasks.

**Questions For Authors:**

Algorithm 1 poses many questions about the method that go unanswered:

1.	What does “Observe()” mean?
2.	What is “Reason()”? Define this and explain how it is implemented.
3.	What is “RequiresUserInput()”? How is this determined?
4.	What is “GenerateUserPrompt()”?
5.	How is it determined whether the agent can generate a response?
6.	How is tool selection performed?

Major questions:

7.	What specifically is lacking about existing benchmarks that required the creation of a new one?
8.	Can the authors provide additional evaluation on individual tasks or subsets of tasks in order to facilitate comparison with more relevant baselines? E.g., might it be possible to form comparisons on select tasks with M4CXR [2], MedVersa [3], or even Google’s MedPalm M [4] if using the same evaluation benchmarks as them? For a method with such diverse capabilities, the extent of evaluation feels underwhelming.
9.	How does MedRAX perform compared to “specialized” models on individual tasks? This could be included as a gray row in each table to provide context for a reasonable upper bound on task performance.

Minor questions:

10.	What does it mean that “all questions underwent a quality check”? Was this performed by anyone with medical expertise?
11.	This is mentioned in the Discussion: “Our initial observations suggest the importance of balanced tool utilization, where neither complete reliance on tools nor their complete absence produced optimal results.” This sounds fascinating but cannot be evaluated since no evidence was provided for this – what observations? Can the authors provide some quantitative (or other) analysis of this in the results?
12.	Were any handcrafted prompts used in evaluation? What were they and how were they decided?

[3] Zhou, Hong-Yu, et al. "A generalist learner for multifaceted medical image interpretation." arXiv preprint arXiv:2405.07988 (2024).

[4] Tu, Tao, et al. "Towards generalist biomedical AI." Nejm Ai 1.3 (2024): AIoa2300138.

**Relation To Broader Scientific Literature:**

This is a unique and ambitious paper that may merit acceptance based on novelty alone. I imagine that this will be among the first of many future works to consider task-agnostic “agentic” approaches to healthcare data analysis, and I commend the authors for their foresight and quick experimentation.

**Theoretical Claims:**

N/A

---

> ### Author Rebuttal · Authors · 2025-04-01
>
> We would like to thank the reviewer for the insightful comments. We have worked hard to answer their concerns and think the suggestions have helped improve the clarity of our work.
>
> **Key Methodoloy.** The full details of Algorithm 1 are provided in response to `Reviewer KTkS`. Additionally, an anonymous GitHub repo is prepared at https://github.com/syaro1383/medrax
>
> **Prior Work.** We have added M4CXR and AgentClinic to the prior work section of the revised manuscript.
>
> **Need for ChestAgentBench.** The justification for making a new benchmark is provided in response to `Reviewer DGwp`.
>
> **Further Evals.** Thank you for suggesting an expanded evaluation with more benchmarks. We've conducted additional benchmarking to compare MedRAX with state-of-the-art models on two benchmarks. Model performances were obtained from Park et al. (2024)
>
> 1. MIMIC-CXR Radiology Report Generation
>
> This benchmark evaluates single-image chest X-ray radiology report generation on the MIMIC-CXR test set, which includes 3,858 images. It assesses the clinical accuracy of generated reports by analyzing the presence or absence of 14 observations of medical conditions using CheXbert.
>
> - mF1-14: Micro-averaged F1 score for all 14 CheXbert observation labels
> - mF1-5: Micro-averaged F1 score for 5 key findings (cardiomegaly, edema, consolidation, atelectasis, pleural effusion)
> - MF1-14: Macro-averaged F1 score for all 14 labels
> - MF1-5: Macro-averaged F1 score for 5 key findings
>
> **Table A: Single-image performance on MIMIC-CXR test set**
>
> | Model | mF1-14 | mF1-5 | MF1-14 | MF1-5 |
> |-------|--------|-------|--------|-------|
> | LLM-CXR† | 36.0 | - | 21.1 | - |
> | RaDialog | - | - | 39.4 | - |
> | METransformer† | - | - | - | - |
> | DCL† | - | - | - | - |
> | PromptMRG | - | - | 38.1 | - |
> | LM-RRG | - | - | - | - |
> | Med-PaLM M 84B | 53.6 | 57.9 | 39.8 | 51.6 |
> | CheXagent* | 39.3 | 41.2 | 24.7 | 34.5 |
> | MAIRA-1 | 55.7 | 56.0 | 38.6 | 47.7 |
> | LLaVA-Rad | 57.3 | 57.4 | 39.5 | 47.7 |
> | M4CXR | 60.6 | 61.8 | 40.0 | 49.5 |
> | MedRAX | 79.1 | 64.9 | 34.2 | 48.2 |
>
> 2. SLAKE VQA Benchmark
>
> The SLAKE benchmark evaluates medical visual question answering using 114 chest X-ray test samples with close-ended questions in English. These questions typically focus on the presence or absence of abnormalities, anatomical identifications, and medical condition assessments.
>
> - Accuracy: Percentage of exact matches between model predictions and ground truth answers
> - Recall: Proportion of ground truth words present in the generated responses
>
> **Table B: Medical VQA performance**
>
> | Model | Accuracy | Recall |
> |-------|----------|--------|
> | RaDialog | 0.0 | 45.6 |
> | RadFM | 68.4 | 69.7 |
> | CheXagent | 71.1 | 73.2 |
> | M4CXR | 85.1 | 86.0 |
> | MedRAX | 90.35 | 91.23 |
>
> **Other Comments.** We have corrected "Llava-Med" to "LLaVA-Med" and added metric descriptions to all table captions.
>
> **Questions.**
> 1-9. Discussed above.
>
> 10. Following the initial generation of questions by GPT-4o based on Eurorad cases, we performed an automated quality verification, also utilizing GPT-4o. Specifically, the automated check evaluated each generated question and answer set for structural consistency (e.g., six-choice question with one correct answer), explicit grounding in the provided clinical and radiological context, and clear verifiability of the correct answer from the original Eurorad source case material. Any questions failing these criteria were automatically identified and excluded. We have provided more details on this verification procedure in the revised manuscript.
>
> 11. During development, we observed that heavily favouring only tool use could lead to rigid or incorrect outputs if a tool failed or misinterpreted, while discouraging tool use missed opportunities for leveraging specialized analysis. Finding a balance, where the agent reasons first but uses tools judiciously to complement and tune its reasoning, appeared to yield better results empirically. A formal quantitative analysis exploring this balance was outside the scope of this paper's experiments but is noted as valuable future work.
>
> 12. Yes, the prompt for MedRAX is as follows:
> > "Answer this question correctly using the chain of thought reasoning and carefully evaluating choices. Solve using your own vision and reasoning and then use tools to complement your reasoning."
>
> This prompt encourages the agent to combine its own reasoning ability and external tools to complement or refine initial assessments. The prompt worked better empirically during development. A formal quantitative comparison of prompting strategies is planned for future work.

---

### Official Review · Reviewer_DGwp · 2025-03-14

**Overall Recommendation:** 3

**Summary:**

The paper introduces MedRAX, an AI-driven agent for the interpretation of chest X-rays (CXRs). MedRAX integrates various specialized state-of-the-art chest X-ray analysis tools and multimodal large language models into a single unified framework. Unlike existing solutions that often operate independently, MedRAX dynamically utilizes these specialized components to handle complex medical queries without additional training.

The authors propose a novel evaluation framework, ChestAgentBench, featuring 2,500 expert-validated questions across seven essential CXR interpretation categories. In comparison experiments, MedRAX significantly outperformed other general-purpose and specialized biomedical models across all assessed tasks, including detection, classification, localization, comparison, relationship understanding, characterization, and diagnosis.

Overall, the study demonstrates that integrating structured reasoning with multimodal specialized tools enhances both accuracy and interpretability in medical imaging tasks, presenting MedRAX as a practical step towards clinical deployment of automated CXR interpretation systems.

**Claims And Evidence:**

The paper clearly presents the MedRAX framework and thoroughly supports its claims with experimental results, demonstrating substantial advantages over existing methods. The key ideas—structured tool-based reasoning, modular tool integration, and comprehensive benchmarking—are explicitly defined and clearly validated by experimental outcomes.

**Essential References Not Discussed:**

n/a

**Experimental Designs Or Analyses:**

The experimental evaluation uses straightforward accuracy metrics (percentage correct answers), which is suitable given the benchmark’s multiple-choice format. The comparisons with existing state-of-the-art general-purpose and biomedical models are clearly defined and implemented using official code and recommended configurations. The evaluation procedure (including handling retries for invalid responses and using regex-based response parsing) appears clearly described and methodologically sound, avoiding ambiguity in how outcomes are measured or interpreted.

I have a minor concern for the benchmark creation: ChestAgentBench and CheXbench:
- The authors describe the ChestAgentBench as containing 2,500 expert-curated questions across seven essential clinical competencies for chest X-ray interpretation. These categories (Detection, Classification, Localization, Comparison, Relationship, Diagnosis, Characterization) comprehensively represent clinically relevant tasks. I understand this is a comprehensive question pool, but how representative are these questions?
- Questions were generated from expert-curated clinical cases (from Eurorad) using GPT-4o, with a clear methodology ensuring that answers are grounded explicitly in the original case descriptions. Any further verification by human experts for benchmark purposes?

**Methods And Evaluation Criteria:**

The methods consist of integrating various specialized medical AI tools (such as visual question answering, segmentation, grounding, report generation, disease classification, and chest X-ray generation) into a modular, structured reasoning framework known as MedRAX. The authors utilize a Reasoning and Acting (ReAct) loop that iteratively breaks down complex medical queries into sequential analytical steps. This systematic approach aligns well with practical clinical workflows in radiology, where interpretation often involves multiple interdependent steps and reasoning based on various specialized analyses.

**Other Comments Or Suggestions:**

see above

**Other Strengths And Weaknesses:**

see above

**Questions For Authors:**

see above

**Relation To Broader Scientific Literature:**

The paper positions itself clearly within broader scientific literature related to artificial intelligence (AI) in medical imaging, specifically chest X-ray (CXR) interpretation, and contributes by building upon several established ideas and addressing previously identified limitations.

**Theoretical Claims:**

The paper provided does not include theoretical proofs or formal algorithmic claims that require rigorous mathematical validation.

---

> ### Author Rebuttal · Authors · 2025-04-01
>
> We would like to thank the reviewer for the insightful feedback.
>
> **ChestAgentBench Representativeness.**
> We appreciate the reviewer's question on benchmark representativeness. We designed ChestAgentBench to be both representative and capable of assessing advanced reasoning, addressing limitations in prior benchmarks:
> * **Addresses Evaluation Gaps in Complex Reasoning:** Existing benchmarks often focus on simpler, single-step VQA or isolated tasks, which are insufficient for evaluating sophisticated AI agents designed for clinical practice. These simpler tasks do not capture the multi-step diagnostic reasoning, evidence integration, and tool use inherent in real-world radiology workflows. ChestAgentBench was explicitly created to fill this critical evaluation gap by presenting complex challenges that require agents to demonstrate deeper, sequential reasoning and integrated analytical capabilities, truly assessing their readiness for clinical application support.
> * **Ensures Broad Clinical Scope and Realistic Data Distribution:** The benchmark's representativeness is supported by its diverse foundation and statistically verified distributions. Derived from 675 real clinical cases, it includes scenarios from various hospital settings and covers a wide spectrum of common and important chest X-ray findings.
>
> **Distribution by Department:**
> * Emergency Room (ER): 19.7%
> * Intensive Care Unit (ICU): 4.9%
> *(Note: The remaining cases originate from other hospital settings, including general wards.)*
>
> **Frequency of Common Chest X-ray Findings:**
> * Mass: 26.3%
> * Effusion: 24.6%
> * Pleural Effusion: 21.5%
> * Consolidation: 21.3%
> * Nodule: 17.2%
> * Calcification: 10.7%
> * Pneumothorax: 7.6%
> * Lymphadenopathy: 7.6%
> * Pneumonia: 7.2%
> * Emphysema: 7.1%
> * Interstitial findings: 6.9%
> * Bronchiectasis: 5.9%
> * Atelectasis: 4.9%
> * Fibrosis: 4.1%
> * Edema: 3.9%
> * Cavitation: 3.9%
> * Fracture: 3.0%
> * Tuberculosis: 2.6%
> * Metastasis: 2.6%
> * Cardiomegaly: 1.5%
>
> This broad distribution across 53 anatomical areas, varied patient demographics, and numerous pathologies ensures agents are tested on a realistic range of clinical challenges.
>
> * **Grounded in Expert Cases and Assesses Integrated Clinical Competencies:** The benchmark's questions are directly derived from and verifiable against detailed findings and expert discussions within 675 authentic, expert-curated clinical cases, ensuring clinical validity and grounding in real-world knowledge. Furthermore, ChestAgentBench systematically evaluates the integration of seven core clinical competencies (such as detection, localization, comparison, relationship analysis, and diagnosis) through complex question types. This design forces agents to demonstrate multifaceted reasoning akin to a clinician synthesizing diverse information, rather than just performing isolated tasks, thereby providing a comprehensive assessment of their true diagnostic reasoning abilities grounded in expert practice.
>
>
> **Benchmark Quality Check.**
> We thank the reviewer for highlighting the need for clarification on our quality check procedure. Following the initial generation of questions by GPT-4o based on Eurorad cases, we performed an automated quality verification, also utilizing GPT-4o. Specifically, the automated check evaluated each generated question and answer set for structural consistency (e.g., six-choice question with one correct answer), explicit grounding in the provided clinical and radiological context, and clear verifiability of the correct answer from the original Eurorad source case material. Any questions failing these criteria were automatically identified and excluded. We have provided more details on this verification procedure in the revised manuscript.

---

### Official Review · Reviewer_b5BD · 2025-03-14

**Overall Recommendation:** 3

**Summary:**

This paper proposes MedRAX, a modular AI agent that integrates specialized chest X-ray (CXR) analysis tools with large language models to perform complex, multi-step medical reasoning. It introduces ChestAgentBench, a large benchmark of 2,500 expert-curated CXR reasoning tasks, to evaluate its performance. Experiments show MedRAX outperforms both general-purpose and specialized models in CXR interpretation, offering improved accuracy and transparency.

**Claims And Evidence:**

Overall, the technical claims are plausible and generally supported by the experiments. However, i would like to see the following:
1. Statistical significance or confidence intervals around the reported accuracy differences.
2. The failure cases of MedRAX (cases where it fails and the potential reason). Also, for the failure cases of the baselines like LLaVA-MED, is MedRAX provide correct predictions. The authors should discuss why MedRAX is correct in those senses.

**Essential References Not Discussed:**

See Relation To Broader Scientific Literature

**Ethical Review Concerns:**

See #8 in Other Strengths And Weaknesses

**Ethical Review Flag:**

Flag this paper for an ethics review.

**Ethics Expertise Needed:**

["Privacy and Security"]

**Experimental Designs Or Analyses:**

I would also encourage an experiment explicitly probing spurious correlations within the pretrained models that MedRAX uses. For instance, it is well documented that pneumothorax classifiers may erroneously learn to rely on chest tubes as a proxy signal, leading to biased predictions. A practical approach would be:

1. Identify Influential Concepts. Use a post-hoc interpretability technique—either a concept-bottleneck model or saliency/heatmap analysis—to pinpoint the concepts or image regions that most strongly influence the model’s predictions.

2. Provide These Influential Concepts to GPT‑4o. Along with the model’s prediction, feed the extracted concepts or saliency annotations into GPT‑4⁰ to ask whether those features are causally relevant or potentially spurious.

3. Assess Biases and Explore Corrective Measures. If GPT‑4o flags a concept (e.g., a chest tube) as non-causal or suspicious, that insight can guide further investigation, such as retraining or debiasing strategies to mitigate reliance on that feature.

Such an experiment could not only uncover hidden biases in the classification tools but also enhance the interpretability of MedRAX’s decision-making process, strengthening confidence in its clinical applicability.

**Methods And Evaluation Criteria:**

1. The authors’ chosen method—combining a large language model (LLM) “agent” (GPT‑4⁰ in their reference implementation) with specialized CXR analysis tools in a ReAct loop—makes sense for the complex, multi-step nature of real radiological queries. The evaluation criteria are primarily classification accuracy (six-choice questions in ChestAgentBench, plus standard VQA accuracy on CheXbench), which is a straightforward metric for correctness.

2. While the choice of multi-choice questions (instead of free-form answers) is somewhat simplifying, it does allow for a consistent, reproducible metric. It would be helpful if the paper described the approach for verifying the “best” single correct answer in each question—especially when complex real-world findings can have nuanced interpretations. In general, the methods and metrics are appropriate and sensible for the problem.

**Other Comments Or Suggestions:**

Please answer my points in Other Strengths And Weaknesses and i will raise the score.

**Other Strengths And Weaknesses:**

1. Lack of prospective or real-world clinical validation beyond curated datasets.
2. No statistical significance or error bars reported.
3. Ablation studies would be helpful: for instance, measuring how MedRAX performs if certain tools are disabled (segmentation vs. classification vs. report-generation, etc.).
4. Does the final response also goes through the LLM (GPT 4o). If so, then there is a potential chance of halucination. Does the author have any idea to reduce it.
5. Certain details is not clear. For ex, i believe the interaction with the LLM needs sepcific prompt to generate thought (Reason(state,M) function in Algo 1) and action. However no mention of them in the paper
6. Does the author provide the CXRs to LLM. Then this method will be expensive. So, i would like to see the cost breakdown of the method consuming the LLMs.
7. Is algorithm 1 is an API? then detailed description needed for the API.
8. Sending CXR and other details, like report to GPT4o can be problematic as they can use these data for training and storing them in their server. The authors shall use these guidelines (https://physionet.org/news/post/gpt-responsible-use) to make sure that these patient data wont be shared to commercial LLMs like GPT4o. So details is necessary on how they send the data through LLM.

**Questions For Authors:**

See Other Strengths And Weaknesses.

**Relation To Broader Scientific Literature:**

The authors situate their work in the broader context of:
- LLM-based agent architectures (ReAct, tool orchestration, etc.).
- Medical VQA and radiology-centric models (CheXagent, LLaVA‑Med, MMedAgent, etc.).
- General multimodal LLM frameworks (GPT‑4⁰ with vision, Segment Anything approaches for image segmentation).

They do a good job contrasting MedRAX with prior domain-specific solutions like CheXagent (which is specialized but not agent-based) and broad frameworks like GPT‑4o (strong reasoning but lacking targeted medical tools). I would also rather add "RaDialog: A Large Vision-Language Model for Radiology Report Generation and Conversational Assistance" to the related work.

**Theoretical Claims:**

NA

---

> ### Author Rebuttal · Authors · 2025-04-01
>
> We want to thank the reviewer for their thorough and thoughtful comments.
>
> **Statistical Significance.** We appreciate the reviewer's point about statistical measures. All experiments were run deterministically (LLMs with temperature 0 and deterministic tools), thus eliminating variability in model outputs for identical inputs. Additionally, since MedRAX integrates pre-trained models without any further training or fine-tuning, there is no randomness from training procedures. This approach ensures full reproducibility and makes traditional confidence intervals less applicable in this context.
>
> **Failure Cases.** For MedRAX failure cases, we have observed instances where the LLM becomes overconfident in its own reasoning and neglects to effectively utilize available tools, leading to incorrect conclusions that could have been corrected with proper tool integration. Regarding baseline failures, models like LLaVA-MED often struggle due to limited training data, resulting in poor generalization to diverse clinical scenarios. MedRAX mitigates this limitation by combining specialized tools with a general-purpose LLM that has broader knowledge and reasoning capabilities. We have included specific examples demonstrating these failure patterns in the revised manuscript.
>
> **Experiment Probing Spurious Correlations.** We appreciate the reviewer's thoughtful suggestion on investigating spurious correlations within the pretrained models used by MedRAX. The concern about biased predictions is addressed through diverse tools that provide cross-validation capabilities. Specifically, MedRAX integrates grounding models that highlight disease regions, allowing the LLM to validate classification predictions against visual evidence from multiple sources. This multi-tool approach helps mitigate reliance on any single potentially biased component. We agree that the reviewer's suggested approach represents a valuable direction for our future work.
>
> **Related Work.** We have added "RaDialog: A Large Vision-Language Model for Radiology Report Generation and Conversational Assistance" to the related work section.
>
> **Strengths and Weaknesses.**
>
> 1. Thank you for raising this important question. MedRAX integrates into clinical workflows as an interactive radiologist copilot. Clinicians can ask questions about CXRs (e.g., for disease classification, localization, VQA) via the user-friendly interface to assist in diagnosis. Its flexible deployment options (local or cloud) and modular design address practical IT and privacy barriers to adoption.
>
> 2. Discussed above.
>
> 3. Table 1 serves as a partial ablation, demonstrating the performance differential between standalone VQA tools (e.g., CheXagent), general-purpose models (GPT-4o), and our integrated MedRAX framework. However, we acknowledge that a more granular ablation study selectively disabling specific tools would provide deeper insights into which tools are most critical for different categories. We will include this analysis in the revised manuscript.
>
> 4. The final response is processed through GPT-4o. Our framework is designed to minimize hallucinations by providing comprehensive context from multiple specialized tools, creating an information-rich environment for the LLM. This multi-tool approach helps ground the model's responses in concrete findings from validated medical AI systems rather than relying solely on the LLM's internal knowledge.
>
> 5. MedRAX utilizes LangChain to manage the reasoning process dynamically. The Reason(state,M) function relies on: (1) an initial system prompt defining the agent's role, available tools with descriptions, and required output structure, and (2) the continuously updated conversation history (memory M) containing past thoughts, actions, and tool results. The agent framework routes execution based on the LLM's output: structured tool calls trigger tool execution, while a final response concludes the process. We have included the agent system prompt and tool descriptions in the revised manuscript.
>
> 6.  MedRAX is flexible: while our evaluation used GPT-4o vision for performance, it can operate without vision LLMs by relying on specialized visual tools. The benchmark evaluation cost using GPT-4o vision involved 2,500 questions with on average 1.85 images per question (~512x512px, ~255 tokens per image) at GPT-4o's input rate of $3.75/1M tokens.
>
> 7. A breakdown of core methodology is provided in response to `Reviewer KTkS`
>
> 8.  Regarding patient data privacy, MedRAX's modular architecture supports locally deployed LLMs to prevent sensitive medical data transmission to external servers. Acknowledging reviewer guidelines, it also supports Azure OpenAI service with appropriate opt-out configurations. This flexibility allows institutions to implement MedRAX compliantly via on-premises deployment or properly configured cloud services, meeting their specific privacy requirements.

---

> > ### Comment · Reviewer_b5BD · 2025-04-02
> >
> > Thanks for the rebuttal.
> > I would like to thank the authors especially for the breakdown of core methodology. However, for reducing hallucination, i dont find the answer to be satisfactory. The authors must take some actions on this in future step. I can provide some papers which can help the authors in this directions:
> > 1. Semantic Consistency-Based Uncertainty Quantification for Factuality in Radiology Report Generation. Wang et al.
> > 2. Direct Preference Optimization for Suppressing Hallucinated Prior Exams in Radiology Report Generation. Banerjee et al.
> > Include this limitation of hallucination in details in the discussion.
> >
> > Also, regarding the privacy, if the authors use GPT-4o API directly, please remove that and add the recommendations by MIMIC-CXR guidelines (use azure endpoint or Vertex from Google) in the final code.
> > Else i am pretty satisfied with the rebuttal and i also read the responses for other reviewers as well. I am raising my score.

---

> > > ### Author Response · Authors · 2025-04-03
> > >
> > > Thank you for your thorough review and thoughtful suggestions that have helped improve our manuscript. We appreciate your concerns regarding hallucination reduction and privacy considerations. We definitely intend to implement your recommended approaches in future work to reduce hallucinations, as this represents a key milestone for deploying MedRAX in clinical settings. The papers you've suggested provide valuable directions for this effort. Regarding privacy concerns, we will add support for Azure OpenAI endpoints with appropriate privacy configurations in our final codebase, ensuring compliance with MIMIC-CXR guidelines for patient data protection. These improvements, along with the detailed ablation studies and failure case analyses, will significantly enhance both the technical rigor and clinical applicability of our work.

---

### Official Review · Reviewer_KTkS · 2025-03-14

**Overall Recommendation:** 3

**Summary:**

The key contributions of this work are:
- MedRAX: An AI framework integrating multiple chest X-ray (CXR) analysis tools without extra training, dynamically orchestrating components for complex medical queries.
- ChestAgentBench: An evaluation framework featuring 2,500 queries across seven categories, built from 675 expert-curated clinical cases, to assess multi-step reasoning in CXR interpretation.
- Performance: MedRAX significantly surpasses general-purpose and biomedical-specific models in complex reasoning tasks, offering transparent workflows.
- Interface: A user-friendly interface enabling flexible deployment from local to cloud solutions, addressing healthcare privacy needs.

This paper emphasizes that structured orchestration of medical AI tools, combining large-scale reasoning and domain-specific knowledge, outperforms purely end-to-end models.


**update after rebuttal**

I’m satisfied with the authors’ response to provide more details about the algorithm and dataset. In my initial review, I had some concerns about the fit of this paper for ICML, as I felt the topic might not attract broad interest. However, after reading the other reviews, I’ve reconsidered my stance. Therefore, I am increasing my score from 2 to 3.

**Claims And Evidence:**

This work proposes an agent-based system for chest X-ray interpretation, integrating domain-specific models as tools with general-purpose LLMs for reasoning. Unlike purely end-to-end approaches, this method leverages existing specialized tools, albeit at the cost of increased inference time. The paper persuasively argues that this combined strategy outperforms E2E models, as structured reasoning tailored to the relatively closed-domain nature of chest X-ray interpretation effectively introduces beneficial inductive biases. Experimental results support this claim.

**Essential References Not Discussed:**

I think the Related Work section does a great job of discussing the limitations of previous studies and clearly highlighting how this work differs from them.

**Experimental Designs Or Analyses:**

To compare existing works, the authors have utilized two established benchmarks and introduced ChestAgentBench. The design of ChestAgentBench is generally logical and reasonable, and the statistical overview of the benchmark dataset is appropriate. However, providing a more detailed distribution of findings across different body regions would enhance the dataset's transparency.

For example, it would be beneficial to present the frequency of the most common chest X-ray findings, such as pleural effusion, cardiomegaly, pulmonary nodules/masses, and others, within the dataset. This would offer a clearer understanding of its composition.

Additionally, the category levels in Figure 4d should be standardized, as the current categorization appears somewhat inconsistent. A more informative approach would be to stratify the dataset as follows:
- Statistics by department: ER, ICU, and general ward
- Findings in chest X-ray: Pleural effusion, cardiomegaly, tuberculosis, nodule/mass, pneumothorax, consolidation, etc.

This refinement would provide a more structured and comprehensive breakdown, improving the interpretability of the dataset.

**Methods And Evaluation Criteria:**

This work focuses on integrating several existing tools; however, it lacks technical details on the implementation of core modules in Algorithm 1, such as Reason, RequiresUserInput, and SelectTool, among others. Providing a more in-depth explanation of these components would strengthen the paper by clarifying how these modules function and contribute to the overall system.

**Other Comments Or Suggestions:**

N/A

**Other Strengths And Weaknesses:**

The presentation of this work is good making it easy to follow, and I enjoyed reading the manuscript. This paper serves as a strong application of machine learning in healthcare, making it a good fit for the Machine Learning for Healthcare track (or similar tracks, if exists). However, for the regular track at ICML, it is unclear whether it would attract significant interest from the broader audience. Since it somewhat lacks novel or particularly compelling machine learning techniques or methodologies that would appeal to the ML research community, the paper's positioning may not align with ICML’s focus.

**Questions For Authors:**

#1. Could you elaborate on why all methods, including MedRAX, perform poorly on image-text reasoning questions? Is it due to the inherent difficulty of this specific task, or because the image datasets originate from external institutions not included during training—essentially representing an external validation scenario common in clinical studies?

#2. The authors argue that MedRAX could be integrated into existing clinical workflows. However, in my understanding, the practical utility and deployment potential of the proposed agent remain unclear. Could you specify how MedRAX could be integrated in daily clinical practice?

#3. Given the low resolution of Figure 2, it's difficult to be certain, but there appears to be an abnormal pattern in both the left and right lungs, possibly indicating bilateral effusions. Therefore, it may be preferable to select a different image for visualization.

**Relation To Broader Scientific Literature:**

This work is highly relevant to foundation models in the medical domain, as it explores an alternative approach to integrating sophisticated existing models with reasoning modules, rather than relying solely on an end-to-end foundation model.

**Theoretical Claims:**

N/A

---

> ### Author Rebuttal · Authors · 2025-04-01
>
> We thank the reviewer for their insightful feedback and appreciate their recognition of the importance of this problem in the healthcare domain.
>
> **Algorithm 1 Core Methodology.** Algorithm 1 outlines the iterative reasoning process of the MedRAX agent in the ReAct loop, as follows:
>
> 1.  **`Observe(Q, I, M)` / State Preparation:** This step initializes the reasoning cycle by gathering the necessary context. It aggregates the user's query (`Q`), any associated input images (`I`), and the entire history maintained in the agent's memory (`M`), which includes previous interactions, tool outputs, and reasoning steps. This consolidated state is fed into the LLM.
>
> 2.  **`Reason(state, M)` -> `thoughts` (LLM Reasoning):** The LLM analyzes the current state (query, images, memory) and available tools (`T`) to generate internal 'thoughts'. These thoughts form a plan deciding the next action: generate a final response, ask the user for input, or select one or more tools.
>
> 3.  **`RequiresUserInput(thoughts)` (Implicit Decision):** This condition is implicitly evaluated by the LLM during the `Reason` step. If the LLM's 'thoughts' indicate ambiguity or insufficient information that cannot be resolved by tools, it will opt to generate a clarifying question for the user instead of proceeding with a tool call or final answer.
>
> 4.  **`GenerateUserPrompt(thoughts, M)` (Implicit Action):** If `RequiresUserInput` is effectively true, the LLM's generated output *is* the prompt for the user. It formulates a natural language question based on its 'thoughts' and the context in memory (`M`) to elicit the needed information.
>
> 5.  **`SelectTool(thoughts, T, M)` -> `tool(s)` (LLM Tool Selection):** When the LLM's 'thoughts' indicate a need for specific capabilities (e.g., classification, segmentation), it selects the most appropriate tool or *multiple tools* from the available set (`T`). This selection is based on the LLM matching its reasoning to the predefined descriptions and capabilities of each tool. For each selected tool, the LLM also formulates the necessary input arguments in a structured format (e.g., JSON).
>
> 6.  **`Execute(tool(s))` -> `result(s)` (Tool Execution):** The agent executes the selected tool function(s). This involves invoking each tool with the structured arguments prepared by the LLM in the `SelectTool` step. The output(s) of these execution(s) are captured as `result(s)`.
>
> 7.  **`M ← M ∪ (thoughts, tool(s), result(s))` (Memory Update):** Following successful tool execution(s), the agent's memory (`M`) is updated. This crucial step logs the LLM's 'thoughts' that led to the tool call(s), the identity of the `tool(s)` used, and the `result(s)` obtained. This ensures that outputs from tools become part of the context for subsequent reasoning cycles.
>
> 8.  **`CanGenerateResponse(thoughts)` / `GenerateResponse(thoughts, M)` (Response Generation):** Following `Reason`, this checks if the LLM's 'thoughts' plan a tool call. If not (`CanGenerateResponse` effectively true), the LLM synthesizes the final natural language answer (`GenerateResponse`), drawing upon its concluding 'thoughts' and information in memory (`M`). If 'thoughts' indicate a tool is needed, this step is bypassed, and the agent proceeds to `SelectTool` and `Execute`.
>
> **ChestAgentBench Statistics.** Thank you for the thoughtful comment. The distribution of common findings and department origins of cases are provided in response to `Reviewer DGwp`. We have revised Figure 3d to incorporate these detailed statistics.
>
> **ICML Relevance.** We submitted MedRAX under the "Application-Driven Machine Learning" track highlighting areas like healthcare. Our work directly addresses significant challenges within this critical domain, and we chose ICML because we believe high-quality, impactful healthcare AI research, while perhaps historically under-represented, is important for the ML community.
>
> **CheXbench Image-Text Reasoning.** This benchmark assesses fine-grained visual reasoning, requiring models to differentiate between options with subtle but critical distinctions (e.g., 'left' vs. 'right' pleural effusion). We observed poor performance across all evaluated models on this task. This suggests the challenge lies in the inherent difficulty of fine-grained radiological interpretation rather than the benchmark dataset itself (derived from the widely-used OpenI).
>
> **Clinical Relevance.** Thank you for raising this important question. MedRAX integrates into clinical workflows as an interactive radiologist copilot. Clinicians can ask questions about CXRs (e.g., for disease classification, localization, VQA) via the user-friendly interface to assist in diagnosis. Its flexible deployment options (local or cloud) and modular design address practical IT and privacy barriers to adoption.
>
> **Figure 2.** We have replaced Figure 2 with a clearer case showing distinct unilateral findings.

---

### Decision · Program_Chairs · 2025-05-01

**Decision:**

Accept (poster)

**Comment:**

This paper introduces MedRAX, a modular medical reasoning agent that integrates multimodal tools and LLMs for chest X-ray interpretation. It also presents ChestAgentBench, a benchmark for evaluating complex clinical queries. The system performs well in comparison to prior generalist and domain-specific models across multiple tasks, and the authors provide a compelling case for its clinical relevance. Reviewers appreciated the novelty and utility of the agent-based architecture, the quality of the benchmark, and the thoughtful design of the evaluation. The rebuttal effectively addressed concerns around implementation detail and evaluation breadth including new results on MIMIC-CXR and SLAKE. Some reviewers increased their scores after these clarifications while one reviewer maintains their weak reject score. AC believes the paper will be valuable for the community and the pros overall outweight the limitations. Therefore the paper is recommended for acceptance. Authors are reminded to improve the implementation details in revision and open-source the code as promised.